# Atlas of plasma NMR biomarkers for health and disease in 118,461 individuals from the UK Biobank

Heli Julkunen [1] ✉, Anna Cichońska[1], Mika Tiainen[1], Harri Koskela[1], Kristian Nybo[1], Valtteri Mäkelä[1], Jussi Nokso-Koivisto[1], Kati Kristiansson [2], Markus Perola[2], Veikko Salomaa [2], Pekka Jousilahti [2], Annamari Lundqvist[2], Antti J. Kangas[1], Pasi Soininen[1], Jeffrey C. Barrett[1] & Peter Würtz[1] ✉

Blood lipids and metabolites are markers of current health and future disease risk. Here, we describe plasma nuclear magnetic resonance (NMR) biomarker data for 118,461 participants in the UK Biobank. The biomarkers cover 249 measures of lipoprotein lipids, fatty acids, and small molecules such as amino acids, ketones, and glycolysis metabolites. We provide an atlas of associations of these biomarkers to prevalence, incidence, and mortality of over 700 common diseases (nightingalehealth.com/atlas). The results reveal a plethora of biomarker associations, including susceptibility to infectious diseases and risk of various cancers, joint disorders, and mental health outcomes, indicating that abundant circulating lipids and metabolites are risk markers beyond cardiometabolic diseases. Clustering analyses indicate similar biomarker association patterns across different disease types, suggesting latent systemic connectivity in the susceptibility to a diverse set of diseases. This work highlights the value of NMR based metabolic biomarker profiling in large biobanks for public health research and translation.

UK Biobank is a prospective study of ~500,000 individuals who have volunteered to have their health information shared with scientists across the globe to advance public health research. This open resource is unique in its size and availability of extensive phenotypic and genomic data[1–3]. A selection of 30 routine blood biomarkers has previously been measured in the full cohort[4,5], but there is a unique opportunity to evaluate the public health relevance of a wider range of biomarkers and accelerating translation, as exemplified by genome-wide genotyping for population-based risk identification[6].

Here, we describe detailed metabolic biomarkers quantified by nuclear magnetic resonance (NMR) spectroscopy of 118,461 baseline plasma samples, generated by Nightingale Health Plc (Fig. 1a). The sample size is more than ten-fold larger than many of the largest metabolic profiling studies conducted to date[7,8]. The NMR biomarker panel comprises 249 measures of lipids and metabolites (Fig. 1b).

These data are now available to approved researchers through the UK Biobank Showcase for all aspects of public health research. Many studies are already using these biomarker data, spanning applications related to, for instance, risk prediction, causal analyses, genetic discovery and drug target validation[9–18].

In this study, we present a comprehensive atlas of biomarker-disease associations (available at nightingalehealth.com/atlas), systematically examined across the 249 metabolic measures in relation to presence, future onset and mortality of over 700 disease outcomes (Fig. 1c). We illustrate the use of the atlas for biomarker discovery and identification of connections between overall biomarker signatures for various diseases. We replicate the findings in over 30,000 individuals from five prospective cohorts in the Finnish Institute for Health and Welfare (THL) Biobank profiled using the same NMR platform. Our biomarker-disease atlas may serve as a starting point to move from

[1]Nightingale Health Plc, Helsinki, Finland. [2]Department of Public Health and Welfare, Finnish Institute for Health and Welfare, Helsinki, Finland. ✉e-mail: Heli.julkunen@nightingalehealth.com; Peter.wurtz@nightingalehealth.com

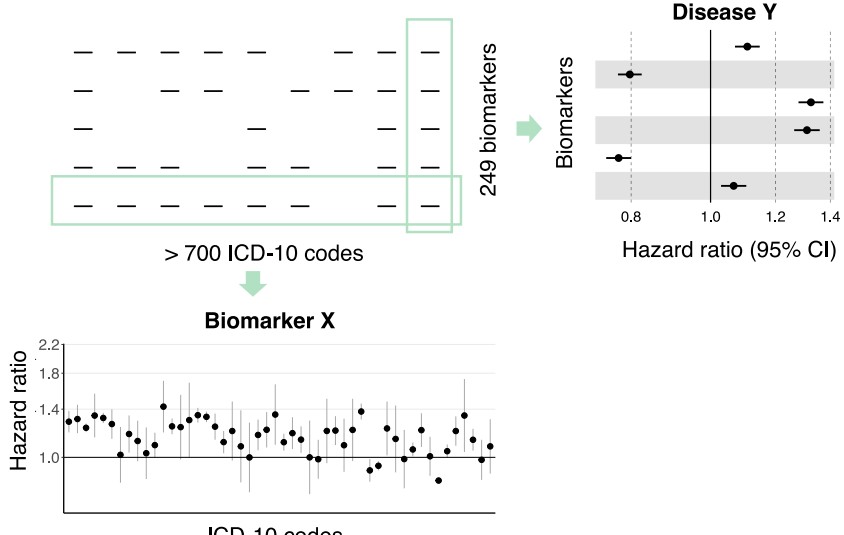

**a PROCESS**

1. Plasma samples sent to Nightingale Health
2. Sample preparation
3. NMR biomarker profiling
4. Biomarker quantification
5. Quality control
6. Data release

**b COMPREHENSIVE BIOMARKER PROFILE**

- Amino acids
- Fatty acids
- Glycolysis metabolites
- Inflammation
- Fluid balance
- Ketone bodies
- Apolipoproteins
- Cholesterol
- Triglycerides
- Phospholipids
- Particle size
- 14 lipoprotein subclasses

**c BIOMARKER-DISEASE ATLAS**

249 biomarkers

> 700 ICD-10 codes

Disease Y — Biomarkers — Hazard ratio (95% CI)

Biomarker X — Hazard ratio — ICD-10 codes

**Fig. 1 | Nuclear magnetic resonance (NMR) biomarker data in the UK Biobank and atlas of disease associations. a** Process of the Nightingale Health-UK Biobank Initiative: 1) EDTA plasma samples from the baseline survey were prepared on 96-well plates and shipped to Nightingale Health laboratories in Finland, 2) Buffer was added and samples transferred to NMR tubes, 3) Samples were measured using six 500 MHz proton NMR spectrometers, 4) Automated spectral processing software was used to quantify 249 biomarker measures from each sample, 5) Quality control metrics based on blind duplicates and internal control samples were used to track consistency metrics throughout the project, 6) Biomarker data were cleaned, provided to UK Biobank and released to the research community. **b** Overview of biomarker types included in the Nightingale Health NMR biomarker panel. **c** Schematic illustration of the atlas of biomarker-disease associations published along with this study. The webtool allows to display the associations of all biomarkers versus prevalence, incidence and mortality of each disease endpoint, as well as show each biomarker versus all disease endpoints.

biomarker discovery to more detailed analyses in biological and clinical context.

## Results

### Plasma biomarker profiling by NMR

We measured lipid and metabolite biomarkers from 118,461 baseline plasma samples using the Nightingale Health NMR platform (Fig. 1a)[7,9,19]. Table 1 shows the characteristics of the participants with NMR biomarker data currently available in the UK Biobank. The EDTA plasma samples were picked randomly and are therefore representative of the 502,543 participants in the full cohort. Samples were generally drawn non-fasting, with an average of 4 hours since the last meal. The data release also contains biomarker measurements of ~4000 repeat visit samples collected on average four years after the baseline, with ~1500 participants having biomarker data from both baseline and the repeat-visit survey.

The Nightingale Health NMR biomarker platform quantifies 249 metabolic measures from each sample in a single experimental assay,

**Table 1 | Characteristics of the UK Biobank participants with plasma NMR biomarkers in the first data release from Nightingale Health**

| | Subset with NMR bio-markers, baseline | Full cohort, baseline |
|---|---|---|
| Number of participants | 118,461 | 502,543 |
| Age at blood sampling (median, [range]) | 58 [39–71] | 58 [37–73] |
| Females (%) | 54 | 54 |
| Body mass index (kg/m²), mean | 27.4 | 27.4 |
| Smoking prevalence (regular, occasional; %) | 7.9, 2.7 | 7.8, 2.7 |
| Fasting time, mean (h) | 3.8 | 3.8 |
| Self-reported cholesterol-lowering medication use (%) | 18 | 17 |

comprising 168 measures in absolute levels and 81 ratio measures (Fig. 1b). The biomarkers include measures already routinely used in clinical practice, such as cholesterol, as well as many emerging biomarkers increasingly measured in cohorts, such omega-3 and other fatty acids[7,20]. The panel of biomarkers is based on feasibility for accurate quantification in a high-throughput manner, and therefore mostly reflects molecules with high circulating concentration. Most of the biomarkers relate to lipoprotein metabolism, with the lipid concentrations and composition measured in 14 lipoprotein subclasses in terms of triglycerides, phospholipids, total cholesterol, cholesterol esters, and free cholesterol, and total lipid concentration within each subclass. The panel additionally includes the absolute concentration and relative balance of the most abundant plasma fatty acids, such as saturated fatty acids, and small molecules, like amino acids, and ketone bodies. Apolipoproteins B and A1, and two inflammatory protein measures, albumin and glycoprotein acetyls, are also measured, owing to their high abundance in plasma.

Details of the NMR biomarker measurements of the UK Biobank samples are described in 'Methods'. Key steps of the measurement process are illustrated in Supplementary Fig. 1 and an overview of all measured biomarkers is provided in Supplementary Fig. 2. The quality control protocol is described in Supplementary Methods and Supplementary Fig. 3. Coefficients of variation of the biomarkers are shown is Supplementary Fig. 4 and technical as well as biological variability illustrated in Supplementary Fig. 5. Comparisons of the NMR biomarker measurements to routine clinical chemistry is illustrated in Supplementary Fig. 6 and to other multi-biomarker assays measured in smaller cohorts in Supplementary Figs. 7 and 8.

## Atlas of biomarker–disease associations

The extensive electronic health records in the UK Biobank and the unprecedented sample size make it possible to study biomarker associations across the full spectrum of common diseases. We systematically computed the associations of the 249 NMR biomarkers with over 700 disease endpoints. Incident and mortality endpoints were defined by 3-character ICD-10 codes from nationwide hospital episode statistics and death records for diseases with at least 50 events occurring during 10 years after blood sampling. Prevalent endpoints were defined for diseases with over 50 events in the hospital records during ~25 years before the blood sampling. Details of the data preprocessing and statistical modelling are described in Methods. We collated the results in form of an online atlas of biomarker-disease associations available at nightingalehealth.com/atlas (Fig. 1c). The webtool can display interactive forestplots for all biomarkers with prevalence, incidence, and mortality of each disease endpoint, as well as disease-wide association plots for each of the 249 biomarkers.

We observed a total of 33,764 individual biomarker associations to incident disease endpoints at $p < 5e-5$ (Methods). Similarly, for 648

prevalent disease endpoints and 77 causes of death, 26,035 and 3,055 significant associations were identified, respectively. These biomarker associations were not concentrated in cardiometabolic diseases but spread across nearly all ICD-10 chapters. Examples include infectious diseases of both systemic and local character, certain cancers as well as mental and neurological disorders and musculoskeletal diseases. The magnitudes of biomarker associations for these diverse types of diseases were often similar to those of cardiovascular diseases. In the subsequent analyses in this paper, we focus on analyses of the future onset of diseases from ICD-10 chapters A-N and the 37 biomarkers from the Nightingale Health NMR platform certified for diagnostic use.

## Biomarkers across the spectrum of diseases

Examining the NMR biomarkers across the spectrum of common diseases can provide insights into disease pathophysiology and specificity of the biomarkers. Fig. 2a illustrates the span of diseases in different ICD-10 chapters associated with the 37 clinically certified biomarkers. Many of the biomarkers exhibited associations across all types of diseases, with the exception of diseases of the eyes and the ears. For example, monounsaturated fatty acids relative to total fatty acids (MUFA%) were associated with almost 200 different disease endpoints spanning all ICD-10 chapters A-N. Also, more established biomarkers such as omega-3% (i.e. concentration relative to total fatty acids) and routine cholesterol measures were associated with a wide spectrum of diseases. Glycolysis-related metabolites and amino acids displayed fewer associations, but still spanned more than endocrine and circulatory diseases.

Figure 2b–e shows the strongest incident disease associations in detail for four exemplar biomarkers; further examples are shown in Supplementary Fig. 9. The inflammatory biomarker glycoprotein acetyls, also known as GlycA, was associated with the risk of 32% of the incident disease endpoints examined ($p < 5e-5$), with a median hazard ratio of 1.26 per 1-SD increment in the biomarker concentration. The most significant associations were observed for gout, type 2 diabetes, smoking dependence, kidney diseases, chronic obstructive pulmonary disorder, myocardial infarction, pneumonia and anemias. Figure 2c highlights the strongest disease associations for the ratio of polyunsaturated fatty acids to monounsaturated fatty acids (PUFA/MUFA), showing as widespread disease associations as for GlycA. Similar results were observed also for other fatty acid measures, such as omega-3% and omega-6% as well as MUFA% (Supplementary Fig. 9a–c).

By contrast to this pattern of diverse associations, some biomarkers exhibited more distinct disease specificity. For instance, the amino acid alanine was primarily associated with the risk of diabetes and complications related to diabetes (Fig. 2d). Glycine and glutamine (Supplementary Fig. 9d, e) were also associated with diabetes-related complications, but additionally with the risk of liver and kidney diseases, with lower plasma concentrations indicating higher disease risk. Glycine was also strongly associated with many circulatory disease endpoints, in line with the earlier suggested causal role of glycine levels in coronary heart disease[21]. Most of the biomarkers had a consistent direction of associations across different diseases, but not all. For example, higher branched-chain amino acid levels were associated with a higher risk for many metabolic diseases but a lower risk for a range of other diseases such as lung diseases, hernia and smoking dependence (Fig. 2e). A small number of biomarkers showed only weak magnitude of association across the spectrum of diseases, such as the ketone body 3-hydroxybutyrate (Supplementary Fig. 9f).

Considered from the disease perspective, Fig. 3 shows the biomarker association profiles for the incidence of six exemplar diseases. Multiple biomarkers are associated with incident hospitalisation for sleep disorders, depression, lung cancer and sepsis, with magnitudes of associations generally similar to those of myocardial infarction. The majority of the biomarkers associated exclusively in one direction of

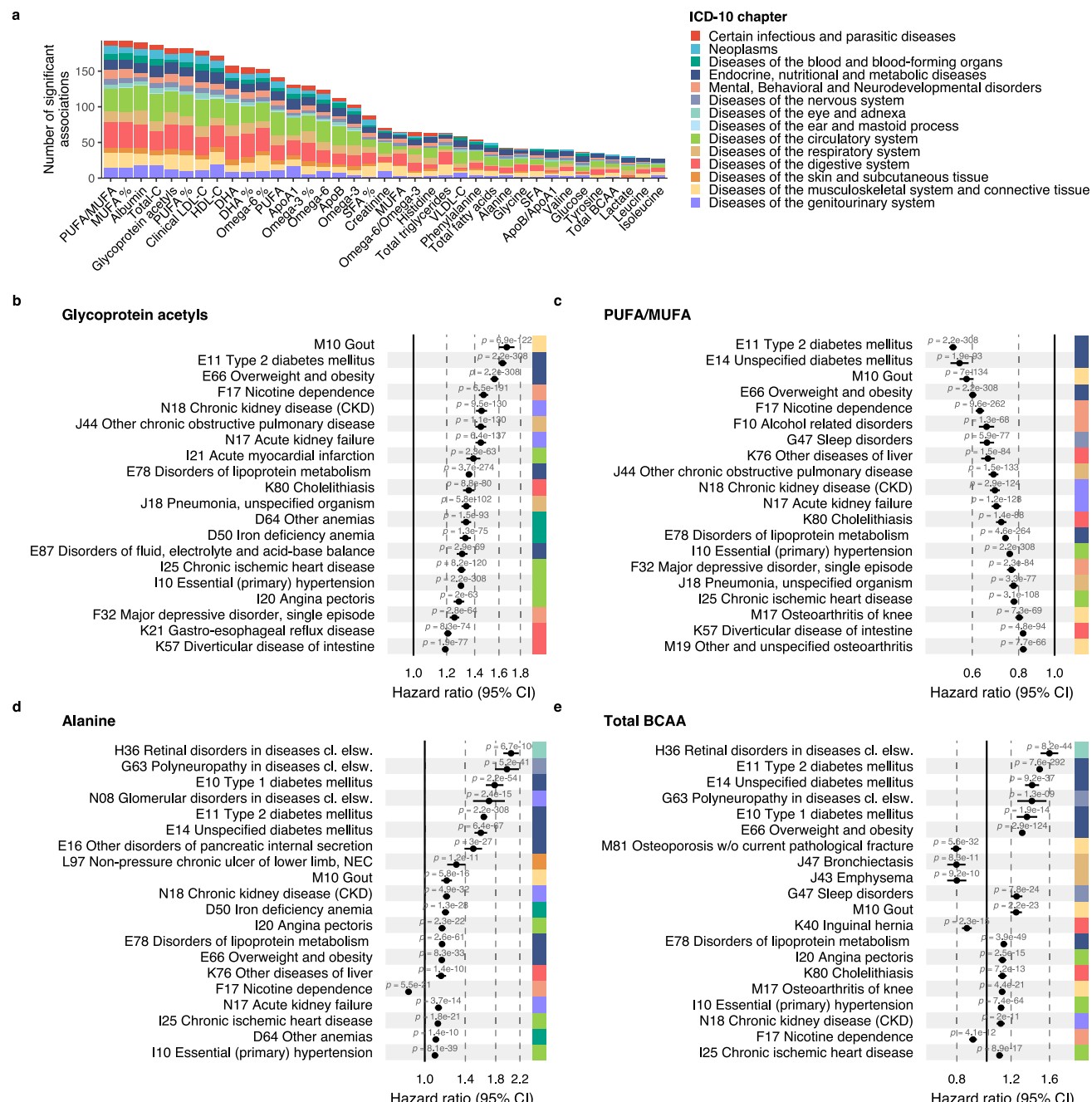

**Fig. 2 | Biomarkers for future disease onset across a spectrum of diseases.**
**a** Total number of incident disease associations by biomarker at statistical significance level $p < 5e\text{-}5$. The disease outcomes were defined based on 3-character ICD-10 codes with 50 or more events from chapters A-N, with a total of 556 diseases tested for association. The colour coding indicates the proportion of associations coming from each ICD-10 chapter from A to N. **b**–**e** Twenty most significant associations for four biomarkers: **b** Glycoprotein acetyls, **c** Ratio of polyunsaturated fatty acids to monounsaturated fatty acids (PUFA/MUFA), **d** Alanine, and

**e** Branched-chain amino acids (BCAA). The forestplots highlight 20 of the most significant associations, arranged according to decreasing association magnitude. Data are presented as hazard ratios and 95% confidence intervals (CI), per SD-scaled biomarker concentrations. All models were adjusted for age, sex and UK biobank assessment centre, using age as the timescale of the Cox proportional hazards regression. Similar disease-wide association plots for all 249 biomarkers across all endpoints analysed are available in the biomarker-disease atlas webtool. Source data are provided as a Source Data file.

effect across these diseases and exhibited similar association patterns overall. An exception to this is osteoporosis, for which increased risk was characterised by decreased concentrations of branched-chain amino acids and triglycerides, and higher high-density lipoprotein cholesterol and apolipoprotein A1—in contrast to the other diseases in Fig. 3. All biomarker associations were robust to a sensitivity analysis excluding the first two years of follow-up, suggesting that they are not driven by clinically incipient cases at baseline (Supplementary Fig. 10).

## Shared biomarker signatures for different diseases

Comparing biomarker signatures between diseases may help to understand molecular differences between conditions with similar pathophysiology and identify novel connections[8,22]. Figure 4a shows examples of clustering of diseases according to their overall biomarker association patterns. In the vertical direction, biomarkers such as GlycA and MUFA% cluster together due to their similarity in associations with many different types of diseases. Most amino acids cluster

**Fig. 3 | Biomarker profiles for the incidence of various types of diseases.** Hazard ratios of biomarkers with the incidence of six disease examples: A41 Sepsis (red; $n$ = 117,806, 2986 events), C34 Lung cancer (light blue; $n$ = 117,964, 1210 events), F32 Depression (green; $n$ = 116,993, 5455 events), G47 Sleep disorders (dark blue; $n$ = 117,325, 1865 events), I21 Myocardial infarction (orange; $n$ = 116,797, 2523 events) and M81 Osteoporosis (lavender; $n$ = 117,538, 3326 events). Data are presented as hazard ratios and 95% confidence intervals (CI), per SD-scaled biomarker concentrations. The models were adjusted for age, sex and UK biobank assessment centre, using age as the timescale of the Cox proportional hazards regression. Filled points indicate statistically significant associations ($p$ < 5e-5), and hollow points are non-significant ones. Similar forest plots for all 249 NMR biomarkers across all endpoints analysed are provided in the biomarker-disease atlas webtool. BCAA indicates branched-chain amino acids, DHA docosahexaenoic acid; MUFA monounsaturated fatty acids, PUFA polyunsaturated fatty acids, SFA saturated fatty acids. Source data are provided as a Source Data file.

together, but glycine and histidine have deviating associations more similar to those of omega-6% and omega-3%, respectively. In the horizontal direction, the clustering analysis reveals both well-known connections between diseases and less anticipated similarities. For example, diabetes has highly similar biomarker association patterns with several of its complications, including polyneuropathies and retinal disorders. Common diseases of an infectious origin, pneumonia and general bacterial infection, also cluster together in terms of their overall biomarker association patterns, as does COPD and lung cancer. Some of the less well-known connections include, for instance, liver diseases and polyneuropathies which had almost identical overall biomarker associations as further highlighted in Fig. 4b.

The biomarker signatures were similar for many diseases, but notable differences may still be observed for diseases of similar

pathophysiological origin[20]. Figure 4c illustrates how acute myocardial infarction and hospitalisation for heart failure have many deviating biomarker associations even though these two endpoints are often combined for clinical trial analyses in the five-point major adverse cardiovascular event (MACE) definition. Supplementary Figs. 11–13 further illustrate similarities and differences in the biomarker signatures for various other types of cardiovascular diseases. The biomarker association pattern differed for different types of myocardial infarction, angina, chronic ischaemic heart disease, and different types of stroke. Even more pronounced differences were observed when compared to heart failure and peripheral artery disease. In particular, many biomarker associations appeared to be stronger for other circulatory endpoints than for myocardial infarction and ischaemic stroke. These results may suggest potential

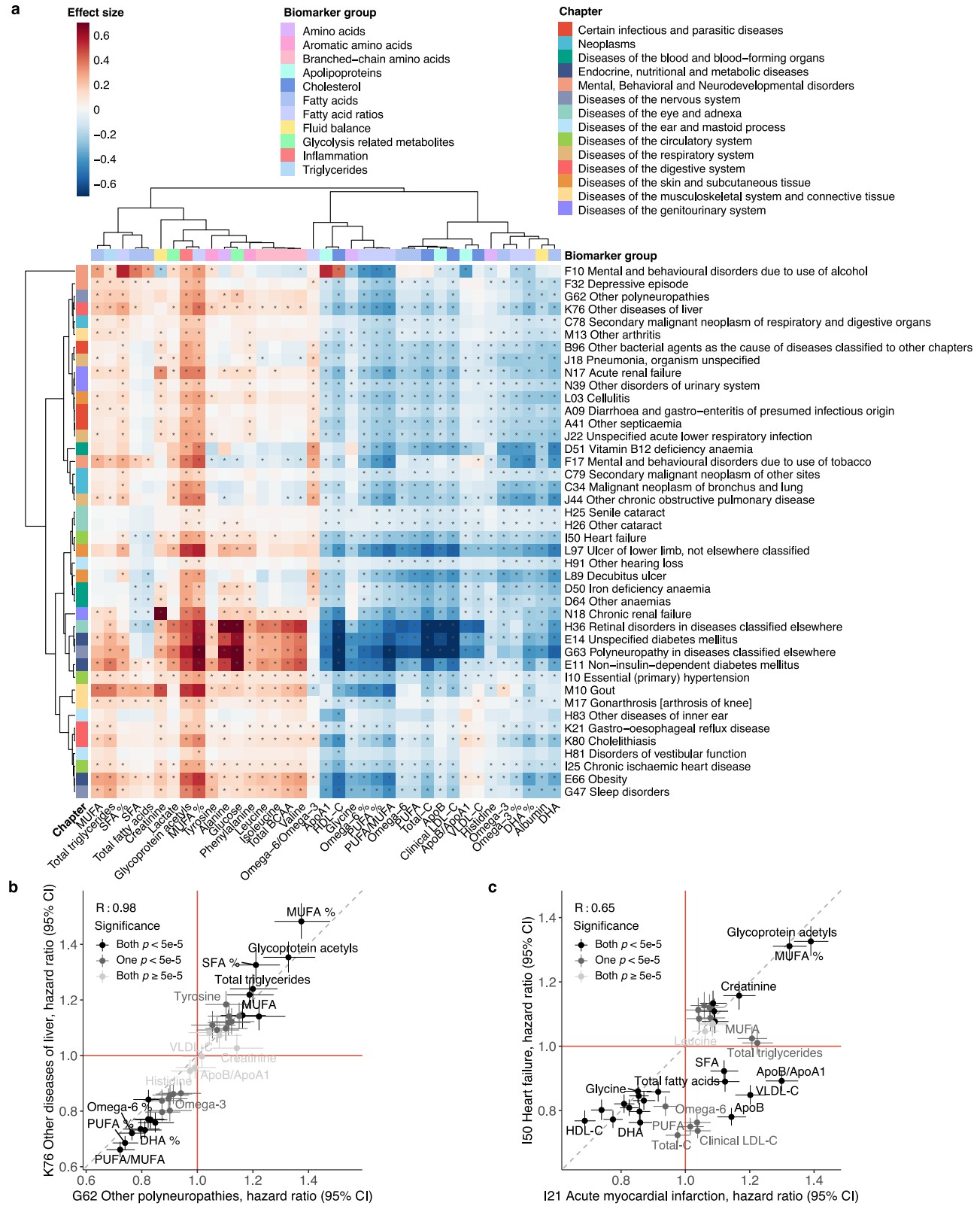

benefits for risk prediction separately for these types of cardiovascular events.

## Replication of biomarker signatures

Replication is essential in biomarker studies, no matter the sample size of the discovery analyses. We, therefore, sought to replicate the NMR biomarker associations in the UK Biobank in two ways: first by comparing the results to biomarkers measured by independent laboratory assays from the same UK Biobank samples, and second by analysing NMR biomarker data for over 30,000 participants from the Finnish Institute for Health and Welfare Biobank (THL biobank). Figure 5 shows the high concordance between disease associations for the eight biomarkers that have been measured by both NMR and clinical chemistry. The associations always have the same direction,

**Fig. 4 | Clustering of incident diseases according to their biomarker signatures. a** Heatmap showing the clustering of biomarker association signatures for the incidence of a diverse set of diseases. The diseases represent three diseases from each ICD-10 chapter from A to N, selected based on the highest number of significant associations. The colouring indicates the association magnitudes in units of the effect sizes, i.e log(hazard ratio per SD). The dendrograms depict the similarity of the association patterns, computed using complete linkage clustering based on the linear correlation between the association signatures. Significant associations with p value < 5e-5 are marked with an asterisk. All models were adjusted for age, sex and UK biobank assessment centre, using age as the timescale of the Cox

proportional hazards regression. Examples of overall biomarker signatures compared for incidence of **b** Other diseases of liver (K76) and Other polyneuropathies (G62), and **c** Acute myocardial infarction (I21) and Heart failure (I50). The hazard ratios for each biomarker are shown as points with 95% confidence intervals (CI) indicated in vertical and horizontal error bars. The colouring of the points indicates the significance of the biomarker association for the pair of diseases. The red lines denote a hazard ratio of 1, and the grey line denotes the diagonal. BCAA indicates branched-chain amino acids, DHA docosahexaenoic acid, MUFA monounsaturated fatty acids, PUFA polyunsaturated fatty acids, SFA saturated fatty acids. Source data are provided as a Source Data file.

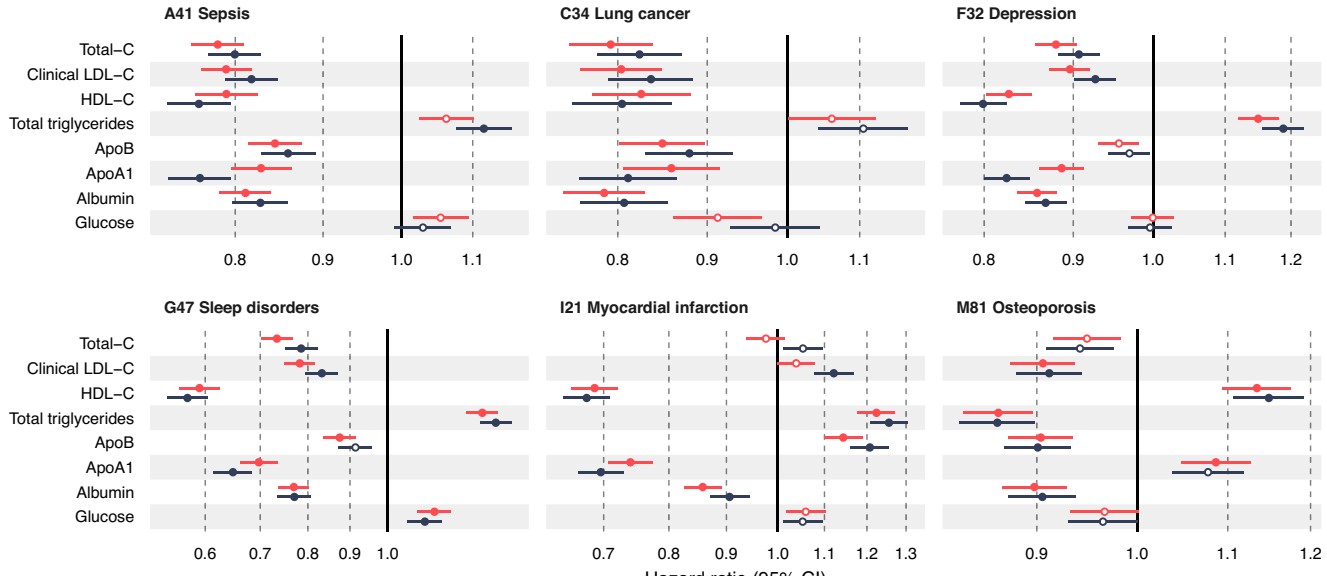

**Fig. 5 | Comparison of nuclear magnetic resonance (NMR) and clinical chemistry biomarker associations.** Hazard ratios of biomarkers for which both NMR-based (red) and clinical chemistry (blue) measurements are available, against the incidence of six disease examples: A41 Sepsis (n = 117,806, 2986 events), C34 Lung cancer (n = 117,964, 1210 events), F32 Depression (n = 116,993, 5455 events), G47 Sleep disorders (n = 117,325, 1865 events), I21 Myocardial infarction (n = 116,797,

2523 events) and M81 Osteoporosis (n = 117,538, 3326 events). Data are presented as hazard ratios and 95% confidence intervals (CI), per SD-scaled biomarker concentrations. The models were adjusted for age, sex and UK biobank assessment centre, using age as the timescale of the Cox proportional hazards regression. Filled points indicate statistically significant (p < 5e-5) associations, hollow points non-significant ones. Source data are provided as a Source Data file.

and the hazard ratios are sometimes stronger for one assay and sometimes another, suggesting neither is systematically better at capturing disease association. Small deviations in the results may be because the plasma samples used for the NMR measurements were more affected by a known sample dilution issue than the corresponding serum samples used for clinical chemistry[4]. The consistency between the NMR-based and clinical chemistry assays in absolute concentrations is illustrated in Supplementary Fig. 7 and further discussed in Methods.

We note that low-density lipoprotein (LDL) cholesterol and apolipoprotein B displayed inverse associations across a wide range of diseases, i.e. higher concentration was associated with lower risk for disease incidence (Fig. 5). This observation, which is surprising compared to the existing literature on LDL as a risk factor for heart disease, is seen in both the NMR and clinical chemistry measurements, indicating that it stems from characteristics of the UK Biobank study rather than any property of the NMR measurements. This observation was mainly explained by widespread use of lipid-lowering medications in the case of cardiovascular endpoints, since the inverse lipid associations were attenuated or inverted direction of effect when individuals on lipid-lowering medication were excluded (Supplementary Fig. 14).

Nonetheless, for most non-circulatory diseases, including five of the six disease examples shown in Fig. 3, the LDL cholesterol associations remained inverse even after excluding individuals on cholesterol-lowering medication (Supplementary Fig. 15), warranting further investigation in other cohorts.

We further replicated the associations observed in UK Biobank by a meta-analysis of five independent population-based cohorts from Finland measured using the same NMR platform (Methods; clinical characteristics listed in Supplementary Table 1). Figure 6 illustrates the consistency of the biomarker association signatures against all-cause mortality and five available incident disease outcomes. Replication results for the remaining available endpoints are shown in Supplementary Fig. 16.

The biomarker associations were generally consistent in the two biobanks, especially for amino acids and other polar metabolites, fatty acid ratios and the two inflammatory protein measures. The results for absolute fatty acid concentrations deviated between the two study populations, whereas the results for fatty acid measures scaled relative to total fatty acids were highly concordant. This may suggest that such ratio measures are more easily transferrable across sampling approaches. The biomarker associations were consistent in

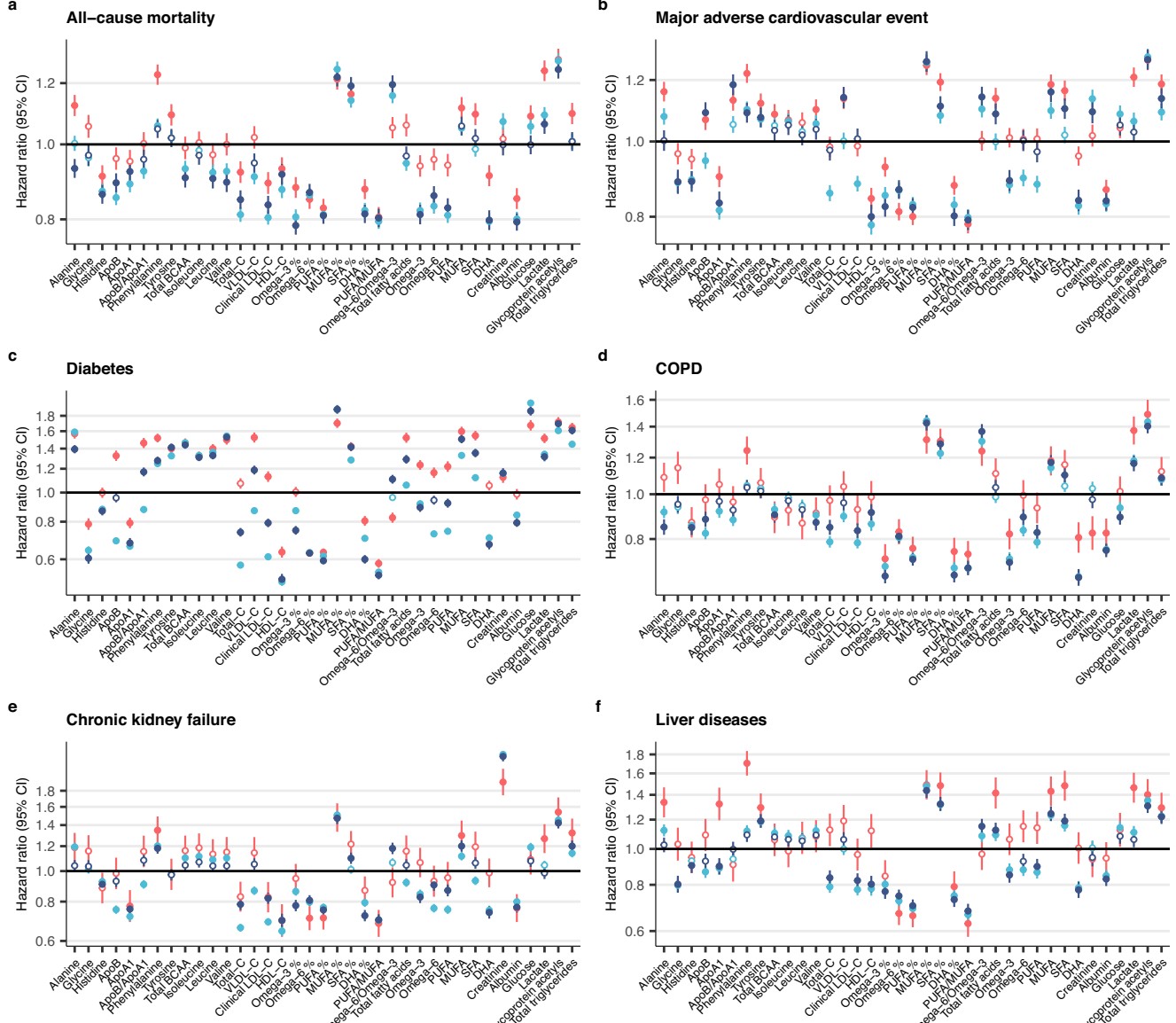

**Fig. 6 | Replication of biomarker associations with incident disease.** Biomarker associations for six disease endpoints are shown for THL Biobank (red) and UK Biobank for the full study population (light blue) as well as for individuals without self-reported use of cholesterol-lowering medication (dark blue): **a** All-cause mortality, **b** Major adverse cardiovascular event, **c** Diabetes, **d** Chronic obstructive pulmonary disease (COPD), **e** Chronic kidney failure and **f** Liver diseases. Results from THL biobank were meta-analysed for five prospective Finnish cohorts (FIN-RISK 1997, 2002, 2007, and 2012, and Health 2000). Data are presented as hazard ratios and 95% confidence intervals (CI), per SD-scaled biomarker concentrations. All models were adjusted for age and sex, using age as the timescale of the Cox proportional hazards regression. Analyses in the UK biobank were additionally adjusted for the UK biobank assessment centre. Filled points indicate statistically significant associations (*p* < 5e-5), and hollow points non-significant ones. Black horizontal line denotes a hazard ratio of 1. Event numbers for incident disease or mortality in the two biobanks are shown in Table 2. ICD-10 codes used for compiling the composite endpoints are listed in Supplementary Table 2. The replication results are shown here for six endpoints available in THL biobank; results for all overlapping endpoints are shown in Supplementary Fig. 16. Results are shown separately for each of the five Finnish cohorts in Supplementary Fig. 17. BCAA indicates branched-chain amino acids, DHA docosahexaenoic acid, MUFA mono-unsaturated fatty acids, PUFA polyunsaturated fatty acids, SFA saturated fatty acids. Source data are provided as a Source Data file.

each of the five Finnish cohorts, although there was a tendency for stronger hazard ratios for the cohort with shortest follow-up time (Supplementary Fig. 17). The greatest deviations were observed for aforementioned LDL-related biomarkers, which displayed strong inverse associations for diabetes and major adverse cardiovascular event (MACE) in UK Biobank but flat or weakly positive associations

in the Finnish cohorts. By excluding participants using cholesterol-lowering medication in the UK Biobank, the associations generally became more consistent (Fig. 6). However, many of the inverse associations for LDL cholesterol and related lipids also replicated in the Finnish cohorts, such as in the case of all-cause mortality and chronic kidney failure (Fig. 6a, e).

**Table 2 | Sample size and number of events for replication analyses**

| Endpoint | THL Biobank Number of events/N (%) | UK Biobank Number of events/N (%) | UK Biobank subset Number of events/N (%) |
|---|---|---|---|
| All-cause mortality | 3 928/34 019 (11.55%) | 7 802/117 868 (6.62%) | 5 219/97 212 (5.37%) |
| Chronic kidney failure | 328/33 982 (0.97%) | 4 254/117 550 (3.62%) | 2 270/97 074 (2.34%) |
| COPD | 732/33 736 (2.17%) | 4 404/117 141 (3.76%) | 2 885/96 811 (2.98%) |
| Liver diseases | 417/33 783 (1.23%) | 2 696/117 328 (2.3%) | 1 884/96 828 (1.95%) |
| MACE | 4 640/31 754 (14.61%) | 6 511/115 745 (5.63%) | 4 311/96 885 (4.45%) |
| Diabetes | 2 703/31 565 (8.56%) | 6 836/115 579 (5.91%) | 3 376/96 746 (3.49%) |

UK Biobank subset represents subset excluding individuals with self-reported use of cholesterol lowering medication.
*COPD* chronic obstructive pulmonary disease, *MACE* major adverse cardiovascular event.

## Age and lipid-lowering medication effects

Excluding individuals using lipid-lowering medication might introduce collider bias in the findings by selecting for healthier individuals. To provide more context for evaluating these results, we also replicated the results in FINRISK 1997 cohort which has a low prevalence of cholesterol-lowering medication use due to the cohort being sampled in 1997 (3.5% in the full cohort, 4.5% after matching age to UK biobank). The results are shown in Supplementary Fig. 18, with analyses matched to the age range of UK Biobank participants. Most of the biomarker associations were consistent in this comparison and the aforementioned inverse and weak associations for LDL-related lipids observed in the UK biobank were also seen in the FINRISK 1997 cohort that is much less affected by cholesterol-lowering medication. This includes, for instance, the null association of LDL cholesterol with MACE and the inverse associations with all-cause mortality and chronic kidney failure. These results suggest that the observations made in UK Biobank after excluding cholesterol-lowering medication users are likely not primarily due to collider bias, but rather relate to the characteristics of the higher-aged individuals in UK Biobank.

To provide another angle on the influence of cholesterol-lowering and other medications on the biomarker associations, we stratified the biomarker analyses by age tertiles[4]. As the use of cholesterol-lowering and other medications increases with age, younger age groups are less prone to such sources of bias. Fig. 7 shows age-stratified biomarker associations for 17 biomarkers across the incidence of the six exemplary diseases from Fig. 3. Results for the remaining 20 biomarkers are shown in Supplementary Fig. 19. In many cases, the association magnitudes were stronger in the youngest age tertile. In particular, notable differences were observed in the case of LDL-related biomarkers, for which the associations became weaker in the older tertiles against myocardial infarction and completely inverted direction against non-circulatory diseases, which can likely be at least partially attributed to the higher prevalence of statin use in the oldest age groups. Increased association magnitudes with younger age were also observed for biomarkers known to not be affected by lipid-lowering treatment[23,24], including inflammatory protein biomarkers and several amino acids, suggesting that the effects cannot be entirely attributed to a lower prevalence of statin use among the younger individuals. Comparison of the age stratified association estimates across all endpoints analysed are available in the biomarker-disease atlas webtool.

## Discussion

Detailed biomarker profiling is a key part of the promise of precision medicine initiatives to transform preventative healthcare. Blood biomarkers provide modifiable molecular measures which relate to future health outcomes and serve as intermediates between lifestyle factors and disease risk. This study describes the generation of NMR biomarker data by Nightingale Health in the UK Biobank, which is currently the world's largest resource of metabolic biomarkers linked to health records. These data greatly extend the blood biomarker coverage in the UK Biobank and provide a wide span of molecular

biomarkers not commonly measured in clinical practice, including amino acids, ketones and fatty acids. With over 118,000 plasma samples profiled in the UK Biobank, the addressable research questions extend vastly beyond biomarker discovery and the large sample size benefits, for example, causal analyses and risk prediction[9,13,14,17]. Due to the streamlined data access policy in UK Biobank, the data release opens possibilities for the research community to use the entire epidemiological toolbox to study the NMR biomarkers in relation to public health.

The biomarkers in the Nightingale Health NMR platform are typically denoted 'metabolic biomarkers', and most prior studies on the data have focused on cardiometabolic diseases. Our analyses reveal that many of these biomarkers capture risk for many other diseases as well. This includes the future onset of diseases of the joints, bones, lungs, many different cancers as well as many mental disorders diseases and severe infectious diseases. These results explain earlier reports on strong associations of the NMR biomarkers with all-cause mortality[25], since many of the biomarkers are associated broadly with leading causes of morbidity and mortality. Widespread associations across different diseases are known for inflammatory biomarkers such as GlycA[26,27], but it has not previously been shown for circulating fatty acids, amino acids or many detailed lipoprotein measures. For example, MUFA% was the biomarker associated across the highest number of endpoints and showed similar disease clustering as GlycA. Our results of widespread disease associations for many fatty acid ratios may suggest that these biomarkers should be considered as markers of systemic inflammation more so than of recent diet.

Plasma metabolites are increasingly understood to link to multimorbidities[8,27]. This is strongly reinforced by our discovery of biomarker associations with the full spectrum of common diseases. We observed that a broad range of diseases with different pathophysiology were characterised by similar biomarker association profiles. For example, severe infectious diseases had similar biomarker signatures to, for instance, chronic respiratory diseases as well as urinary and renal diseases. A potential explanation may be that many of the biomarkers reflect the innate immune system's ability to respond. This would help to explain why many of the biomarkers were associated with susceptibility to severe infectious diseases, such as hospitalisation and death from sepsis, fungal infections and pneumonia[9]. These observations illustrate how novel insights beyond individual diseases can be gained by studying overall biomarker signatures and numerous disease outcomes simultaneously. The genomic data in UK Biobank may help to elucidate causality of these results via Mendelian randomisation[11,17].

The striking similarity of the biomarker risk profiles across various diseases might pose challenges to certain clinical applications requiring high disease specificity. However, it is ideal when aiming to use the biomarker panel to assess the risk of multiple diseases and overall health status simultaneously based on a single measurement. This could potentially be used for individualised health assessment at scale to prioritise high-risk individuals for further examinations and guide

**Fig. 7 | Age-stratified biomarker profiles for the onset of various types of diseases.** Biomarker profiles stratified by age tertiles: 1st tertile (3–53 years of age; dark blue), 2nd tertile (54–61 years of age; red) and 3rd tertile (62–71 years of age; green). Results are shown for 17 biomarkers across six disease examples: A41 Sepsis (*n* = 117,806, 2986 events), C34 Lung cancer (*n* = 117,964, 1210 events), F32 Depression (*n* = 116,993, 5455 events), G47 Sleep disorders (n = 117,325, 1865 events), I21 Myocardial infarction (*n* = 116,797, 2523 events) and M81 Osteoporosis (*n* = 117,538, 3326 events). Results for the remaining 20 biomarkers are shown in Supplementary Fig. 19. Data are presented as hazard ratios and 95% confidence

intervals (CI), per SD-scaled biomarker concentrations. The models were adjusted for age, sex and UK biobank assessment centre, using age as the timescale of the Cox proportional hazards regression. Filled points indicate statistically significant associations (p < 5e-5), and hollow points non-significant ones. Similar forest plots for all 249 NMR biomarkers across all endpoints analysed are provided in the biomarker-disease atlas webtool. DHA indicates docosahexaenoic acid, MUFA monounsaturated fatty acids, PUFA polyunsaturated fatty acids, SFA saturated fatty acids. Source data are provided as a Source Data file.

preventative actions. In fact, a recently published study[28] demonstrated the potential of the NMR biomarker profiles to predict multi-disease outcomes, showing predictive improvements over comprehensive clinical risk factors which were largely shown to translate into clinical utility. As such, this could have many applications in clinical settings and provide an attractive tool for multi-disease risk screening.

Our biomarker-disease atlas published with this paper can be used to rapidly corroborate or refute many prior biomarker studies. For instance, we replicate the recent reports on higher branched-chain amino acid concentrations associated with lower risk for Alzheimer's disease and dementia[29]. The event numbers for these neurodegenerative diseases in UK Biobank alone are similar to those in the meta-analysed eight cohorts. The biomarker-disease atlas may also be used to put into question other reported biomarker discoveries, such as branched-chained amino acids in relation to risk for pancreatic cancer:[30] the association was essentially flat in UK Biobank despite a similar number of events. These examples illustrate how the biomarker-disease atlas may speed up research and serve as a starting point for analyses that yield deeper aetiological insights and clinical context, much as widely available GWAS summary statistics transformed the interpretation of genetic studies. We note that the availability of the NMR biomarker data in UK Biobank does not diminish the relevance of having these data in smaller cohorts, both for replication and for complementary study designs. For example, the precise estimates of biomarker associations in UK Biobank can make analyses of smaller cohorts and trials more interpretable in relation to longitudinal sampling and intervention effects.

Metabolic profiling of all 500,000 baseline plasma samples in UK Biobank is underway. This will greatly expand the possibilities for studying rarer diseases and prediction of short-term risk, as well as open possibilities for analyses focusing on individuals with prevalent disease and multi-morbidity trajectories. Coupled with the rich genomic data, clinical chemistry and proteomics measures, imaging, complete health-records, and other health-related data that are continually added to the UK Biobank resource, the NMR biomarker data will enhance the possibilities for scientific discovery and is set to yield important findings for public health and clinical use. The data are available to approved researchers through similar access protocols as existing UK Biobank data (http://ukbiobank.ac.uk/).

## Methods
### UK Biobank cohort
The UK Biobank study was approved by the North West Multi-Centre Research Ethics Committee and all participants provided written informed consent. The study protocol is available online (https://www.ukbiobank.ac.uk). The biomarker profiling of plasma samples by NMR spectroscopy was approved under UK Biobank Project 30418.

The UK Biobank resource is a globally accessible biomedical database of half a million UK participants aged 40–69 years at baseline[1]. Baseline characteristics of the full cohort and the subset with available NMR biomarker data are provided in Table 1. A large variety of health information has been collected for each participant. For instance, the database includes questionnaire data on participant's socio-economic and lifestyle factors, cognitive tests, imaging data, heart and lung function measures, body size and composition measures. Extensive genomic data is available, with genotyping array and exome-sequencing data available for all participants, and whole-genome sequencing under way[2].

The UK Biobank blood sample collection was undertaken at baseline in 22 local assessment centres across the UK between 2007 and 2010. The blood sample handling and storage protocol has been previously described[31]. Prior to the measurement of the NMR biomarkers, 35 biomarkers have been measured from blood and urine samples by clinical chemistry[4,5].

### Plasma biomarker profiling by NMR
Nightingale Health Plc. is performing biomarker profiling of baseline plasma samples for all 500,000 participants in the UK Biobank. Details of the Nightingale Health NMR biomarker platform have been described previously[7,19]. The main steps in the experimental procedures are illustrated in Supplementary Fig. 1. The biomarker measurements took place in Finland between 2019 and 2020 using six NMR spectrometers. The first data release covers biomarker measurements from a random selection of 118,461 EDTA plasma samples from the baseline recruitment. In addition, around 4000 EDTA plasma samples from repeat assessments are included in the same data release, with both baseline and repeat-visit sample measured for -1500 participants. The NMR biomarker dataset has been made available for the research community through the UK Biobank in March 2021.

All sample analysis processes were performed according to the standard operating procedures that are part of Nightingale Health's EN ISO 13485 certified Quality Management System (certified by DEKRA Certification B.V. Nightingale Health measured all plasma samples with a CE-marked In Vitro Diagnostic Medical Device. At time of completion of UK Biobank phase 1 samples, 37 of the biomarkers in the panel were CE-marked and certified for diagnostics use. In order to facilitate translational applications and visualisation of the results, we focused on this set of 37 clinically validated biomarkers in the examples highlighted in the paper, as they span most of the different metabolic pathways measured by the NMR platform. Complete results for all 249 biomarkers measured are provided in the biomarker atlas webtool.

**Plasma sample preparation.** EDTA plasma samples from aliquot 3 were prepared in 96-well plates by UK Biobank laboratory (Stockport, UK). At least 90 μL of plasma was aliquoted in each well using TECAN freedom EVO 150 robotic liquid handlers, which have coefficients of variation (CV) in pipetting volume at <0.75% across 8 tips. The plasma samples were shipped to Nightingale Health laboratories in Finland in 96-well plates on dry ice in batches of 5000–20,000 samples. No selection criteria were applied to the sampling and the 118,461 samples are therefore a random subset of the full cohort.

Samples were stored in a freezer at −80 °C at Nightingale Health laboratories after arrival from UK Biobank laboratory. Before preparation, frozen samples were slowly thawed at +4 °C overnight, and then mixed gently and centrifuged (3 min, 3400 × $g$, +4 °C) to remove possible precipitate. Aliquots of each sample were transferred into 3-mm outer-diameter NMR tubes and mixed in 1:1 ratio with a phosphate buffer (75 mM $Na_2HPO_4$ in 80%/20% $H_2O/D_2O$, pH 7.4, including also 0.08% sodium 3-(trimethylsilyl) propionate-2,2,3,3-d4 and 0.04% sodium azide) automatically with an automated liquid handler (PerkinElmer Janus Automated Workstation).

**NMR spectroscopy.** The plasma samples were measured using six 500 MHz NMR spectrometers (Bruker AVANCE IIIHD). Measurements were conducted blinded prior to the linkage to the UK Biobank health outcomes. The prepared plasma samples on 96-well plates were loaded onto a cooled sample changer, which maintains the temperature of samples waiting to be measured at +6 °C. Two NMR spectra were recorded for each plasma sample. The first spectrum is a presaturated proton spectrum, which features resonances arising mainly from proteins and lipids within various lipoprotein particles. The second spectrum is a Carr-Purcell-Meiboom-Gill $T_2$-relaxation-filtered spectrum where most of the broad macromolecule and lipoprotein lipid signals are suppressed, leading to enhanced detection of low-molecular-weight metabolites.

**Quantified biomarkers.** The biomarkers were quantified using Nightingale Health's proprietary software (quantification library 2020), which simultaneously quantifies 249 metabolic measures per EDTA plasma sample, comprising 168 absolute and 81 ratio measures

(Supplementary Fig. 2). All the biomarkers are of known-identity. The biomarker measures include routine lipids, lipoprotein subclass profiling with lipid concentrations within 14 subclasses, fatty acid composition, and various low-molecular-weight metabolites such as amino acids, ketone bodies and glycolysis metabolites quantified in molar concentration units. For 14 lipoprotein subclasses, the lipid concentrations and composition are measured in terms of triglycerides, phospholipids, total cholesterol, cholesterol esters, and free cholesterol, and total lipid concentration within each subclass. The majority of the biomarkers are measured in absolute concentration units (mmol/L). The 37 biomarkers in the panel which have been certified for diagnostics use (CE-marked) are marked by asterisks in Supplementary Fig. 2. The average biomarker detection rate was >99% across the plasma samples.

The quality control protocol is described in Supplementary Methods and illustrated in Supplementary Fig. 3. The distribution of coefficients of variation of the biomarkers for UK Biobank's blind duplicate samples as well as Nightingale Health's internal control samples is shown in Supplementary Fig. 4. The coefficients of variation for each biomarker is given in the UK Biobank data resource (https://biobank.ndph.ox.ac.uk/showcase/label.cgi?id=220). This resource also contains distribution plots showing the consistency over consecutive shipment batches and in different NMR spectrometers, as well as scatter plots on the technical repeatability from blinded duplicate samples and the biological consistency in repeat-visit samples drawn from the same individuals four years apart. These technical and biological repeatability assessments are illustrated with GlycA as an example in Supplementary Fig. 5. Supplementary Methods further contain notes about the quality flags for samples and biomarkers as well as general recommendations for data processing in relation to epidemiological analyses.

**Plasma sample dilution issue.** All UK Biobank blood samples are known to suffer from unintended dilution during the initial sample storage process at UK Biobank facilities. Prior reports have suggested that samples from aliquot 3, used for the NMR measurements, suffer from 5-10% dilution[4]. The dilution is believed to come from mixing of participant samples with water due to seals that failed to hold a system vacuum in the automated liquid handling systems. While this issue is likely to have an impact on some of the absolute biomarker concentration values, it is expected to have limited impact on most epidemiological analyses. However, we recommend that this aspect is considered when conducting analyses that rely on absolute concentrations, such as stratification based on biomarker concentration cutpoints. This may also cause challenges to compare distributions of biomarker concentrations with those observed in other cohort studies. We, therefore, caution against using the concentrations observed in UK Biobank as reference levels for translational applications.

**Comparison to clinical chemistry.** The consistency between lipids, apolipoproteins, creatinine, albumin and glucose measured by routine clinical chemistry and Nightingale Health NMR is illustrated in Supplementary Fig. 6. For these comparisons, it is important to note that the clinical chemistry in UK Biobank was measured from serum samples, primarily from aliquot 1, while the NMR biomarkers were measured from EDTA plasma samples from aliquot 3. The different aliquots are affected by different degrees of dilution, with aliquot 3 being 5–10% diluted while aliquot 1 has almost no dilution[4]. Supplementary Fig. 6 therefore also shows the measurement consistency in the FinHealth 2017 study, without the dilution issue. This study is a population-based cohort under the Finnish Institute for Health and Welfare (THL) Biobank with $n \approx 6000$. In the FinHealth 2017 cohort, clinical chemistry assays were measured from frozen serum samples soon after the cohort survey and the NMR biomarkers one year later

from frozen samples using the Nightingale Health platform on 350 μL aliquots of serum.

Correlations between the clinical chemistry assays and NMR were high in both cohorts, but the overall consistency was weaker in UK Biobank compared with the FinHealth 2017 study. In particular, the absolute concentrations were deviating more from the diagonal in UK Biobank in than in the FinHealth 2017 study, owing to the sample dilution issue in UK Biobank. Other aspects contributing to mismatch in absolute concentrations in UK Biobank are subtle differences in biomarker levels between serum and EDTA plasma and longer differences in sample storage time. The consistency of the NMR biomarkers with clinical chemistry in the FinHealth 2017 study is in line with earlier studies that have reported correlation coefficients $R > 0.9$[2]. A recent paper reported correlations of the same NMR biomarkers with clinical chemistry for FINRISK cohorts under the THL Biobank to be R-0.95 for the newest sample collection, and R-0.90 for the oldest sample collections[20]. Note that 'Clinical LDL cholesterol' is the NMR-based measure that provides concentrations consistent with clinical chemistry and the Friedewald equation for LDL-cholesterol. We further note that the correlation coefficient for albumin was weaker in the UK Biobank than observed for the other clinical chemistry measures. However, the associations of albumin with disease outcomes were broadly similar for albumin for both assays as shown in Fig. 5.

Comparisons of the NMR biomarkers with overlapping biomarkers from commercial mass-spectrometry assays and gas chromatography fatty acid assays in smaller cohorts are described in Supplementary Methods and scatter plots of the consistency illustrated in Supplementary Figs. 7 and 8.

**Disease outcome definitions**
Prevalent, incident and mortality disease outcomes were derived from UK Hospital Episode Statistics data and national death registries. A diagnosis in hospital or death record formed the basis of the disease endpoint definition. Primary care records were not used. Disease endpoints were defined based on the first occurrence of 3-character ICD-10 code using the hospital inpatient and death register data (January 2021 update). To extend the follow-time prior to the introduction of ICD-10 in 1995, ICD-9 codes were mapped to the corresponding 3-character ICD-10 codes using general equivalence mappings from Center for Disease Control (https://ftp.cdc.gov/pub/Health_Statistics/NCHS/Publications/ICD10CM/2018/).

A prevalent event was defined as an event that occurred before the date of participant's baseline visit when a blood sample was collected. Individuals with corresponding prevalent event for each outcome were excluded from the analysis of incident disease, but not for analyses of mortality outcomes. The occurrence of both primary and secondary diagnoses codes was considered to form the endpoints. The follow-up of hospitalisations ended on November 30, 2020 in England, October 31, 2020 in Scotland, and February 28, 2018 in Wales. The follow-up of death registry ended on November 30, 2020. We omitted disease outcomes with fewer than 50 cases from the analyses. This led to a total of 648 prevalent, 717 incident and 77 mortality outcomes for the study population with NMR biomarker data available.

For the examples highlighted in this paper, we focused on 556 incident disease outcomes from ICD-10 chapters A-N. The selection of chapters A-N excludes pregnancy-related outcomes, conditions originating in the perinatal period and congenital malformations, deformations and chromosomal abnormalities (chapters O-Q) as there were not enough incident events passing the criteria of over 50 events to be included in the analyses. Chapters R-U (symptoms, signs and laboratory findings not elsewhere classified, injuries, accidents and factors influencing health status and contact with health care services and codes for special purposes) were excluded to place the focus on common diseases.

## Biomarker association analyses across all endpoints

For the disease association analyses, biomarker values outside four interquartile ranges from median were considered outliers and excluded from the analyses. Furthermore, biomarker values were corrected for the NMR spectrometer used for the measurements by fitting a linear regression model with log1p-transformed concentrations as the outcome and spectrometer as the predictor. Scaled residuals from this regression were used as predictors in the association analyses. Log1p stands for the natural logarithm of $1 + x$.

We used Cox proportional hazard modelling to estimate associations between biomarkers and incident disease outcomes (hospitalisation or death) across all endpoints with 50 or more events. The models were adjusted for sex and UK biobank assessment centre, using age as the time scale of the Cox proportional hazards regression. Associations for each biomarker-disease pair were computed separately. For biomarker association testing with prevalent diseases, we used logistic regression models adjusted for age, sex and assessment centre. Hazard ratios and odds ratios are reported per SD increment in the log1p-transformed biomarker concentrations in order to allow comparison of association magnitudes for measures with different units and concentration range. Sex-specific analyses were conducted for 148 female-specific and 18 male-specific diseases (Supplementary Table 3). These association analyses were performed in a subset containing only the specific sex, using the same approach without the inclusion of sex as a covariate. We also performed analyses by stratifying the UK biobank population into age tertiles (1st tertile 39-53 years of age, statin use 6%; 2nd tertile 54-61 years of age, statin use 17%; 3rd tertile 62-71 years of age, statin use 30%).

In the biomarker-disease atlas, results are reported for all conducted analyses and the webtool allows to filter by a desired significance level. In this paper, we use a multiple testing-corrected significance level of $5 \times 10^{-5}$ for reporting statistically significant associations, i.e. correcting for 1000 independent tests to account for both high correlation between the NMR biomarkers (~50 independent tests[7]) and correlations between the disease endpoints analysed.

## Clustering analyses

For clustering analyses, a dendrogram and heatmap were computed based on the association magnitudes of the 37 biomarkers with three diseases from each ICD-10 chapter from A to N. The diseases were selected based on the highest number of significant biomarker associations in each ICD-10 chapter. The 37 biomarkers selected are the ones clinically validated in the Nightingale Health NMR platform. Biomarkers are clustered in the dendrogram based on disease association profiles, and diseases are clustered based on biomarker profiles, using complete linkage clustering based on linear correlation between the association signatures.

## Replication in additional cohorts

To replicate biomarker associations from the UK Biobank, we used data from five prospective population-based studies administered under the Finnish Institute for Health and Welfare (THL) Biobank: FINRISK 1997, FINRISK 2002, FINRISK 2007, FINRISK 2012 and Health 2000. Each cohort is an independent random sample drawn from people aged 25-98 (25–74 in FINRISK, 30 and over in Health 2000) in the Finnish population. Baseline characteristics of these cohorts are provided in Supplementary Table 1. The study participants are unique in each cohort. Baseline blood samples were collected for ~85% of all participants enroled. Venous blood was drawn non-fasting, but with recommended minimum of 4-h fast. Biomarker profiling by the Nightingale Health NMR platform was conducted from frozen serum samples for all participants during 2018[20]. The cohort studies were approved by the Coordinating Ethical Committee of the Helsinki and Uusimaa Hospital District, Finland. Written informed consent was obtained from all participants.

Fourteen disease endpoints were used for replication analyses in THL Biobank, selected based on the outcome data made available to Nightingale Health Plc. The disease outcome definitions were pre-defined by THL Biobank based on a combination of national hospital and cause-of-death registries (Supplementary Table 2). The registry-based follow-up cover virtually all diseases leading to hospitalisation or death in Finland. Follow-up data for the present study were until the end of 2016. For the replication analyses, we defined similar endpoints in UK Biobank based on the ICD-10 codes listed in Supplementary Table 2.

The association analyses were for incident disease, so individuals with prevalent disease of the same endpoint were omitted. The hazard ratios were computed separately in each cohort using Cox proportional hazards regression adjusted for sex and using age as the time scale of the regression. Results from the individual cohorts were meta-analysed using inverse variance weighting. Similar to the analyses in UK Biobank, hazard ratios are reported in SD-scaled units.

## Reporting summary

Further information on research design is available in the Nature Portfolio Reporting Summary linked to this article.

## Data availability

The Nightingale Health NMR biomarker data have been released to the UK Biobank resource in spring 2021 (https://biobank.ndph.ox.ac.uk/showcase/label.cgi?id=220). The UK Biobank data are available for approved researchers through the UK Biobank data-access protocol. NMR spectral data are not available as they are outside of the scope of the Nightingale-UK Biobank initiative. Instructions for the data access process, timeframe and restrictions imposed on the data are described at https://www.ukbiobank.ac.uk/enable-your-research/apply-for-access. The average number of weeks from application submission to data release is 15 weeks for UK Biobank. Nightingale Health NMR biomarker data from FINRISK and Health 2000 cohorts, used for replication in this study, are available for approved researchers through THL Biobank. Instructions for the data access process is provided at https://thl.fi/en/web/thl-biobank/for-researchers/application-process. We provide access to all biomarker-disease summary statistics for non-commercial use through an interactive webtool https://nightingalehealth.com/atlas (CCBY-NC-ND 4.0 license). Source data are provided with this paper.

## Code availability

Code used in this study is available at: https://github.com/NightingaleHealth/ukb-nightingale-biomarker-atlas. Analyses were performed using R (completed and tested with version 4.1.1).

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

## Acknowledgements

The authors are grateful to UK Biobank (Project #30418) and THL Biobank (project #BB2016_86) for access to data to undertake this study. The authors thank all biobank participants for their generous contribution to generating this resource for the scientific community. The work was funded by Nightingale Health Plc.

## Author contributions

H.J., A.C., A.J.K., P.S., J.B., P.W. designed research; H.J. and A.C. contributed to statistical analyses and interpretation of results; M.T., H.K., K.N., V.M., J.N.-K., P.S., A.J.K. contributed to biomarker measurements and quality control; M.P., V.S., P.J., A.L., and K.K. contributed data or results for replication; H.J., A.C., J.B., and P.W. contributed to the interpretation of results and wrote the manuscript. All authors reviewed the manuscript.

## Competing interests

H.J., M.T., H.K., K.N., V.M., J.N.-K., A.J.K., P.S., J.B., and P.W. are employees of Nightingale Health Plc, and hold shares or stock options in Nightingale Health Plc. A.C. is former employee of Nightingale Health Plc. V.S. has received an honorarium for consulting from Sanofi and has ongoing research collaboration with Bayer Ltd outside this work. The remaining authors declare no competing interests.
