## [Peer Review File · Nature Communications]

REVIEWER COMMENTS

Reviewer #1 (Remarks to the Author):

This paper by Julkunen et al presents rich resource of associations between 249 circulating metabolic traits and over 700 ICD10 defined endpoints using data on 118,461 participants from the UK Biobank (UKB) study. As the authors highlight, this sample size provides a 10-fold increase in sample over previous profiling studies to date and further enhances the already amazing phenotypic breadth of the UKB.

This resource paper is an excellent contribution to the field with appropriate statistical analyses, clear presentation and summary of key findings/figures and an accompanying web resource to query/download results. My only major concern is the inherent challenge of conducting NMR profiling in a biobank-scale dataset such as UKB and the most appropriate manner to account for medication usage (in particular statins). The authors touch on this on line 275:

'This observation, which is surprising compared to existing literature on LDL as a risk factor, is seen in both the NMR and clinical chemistry measurements, indicating that it stems from characteristics of the UK Biobank study rather than any property of the NMR measurements.'

The authors conduct further analyses excluding reported statin users from their sample in an attempt to correct for this source of bias, although I'm not sure this is appropriate given that such an approach may induce collider bias into their findings. For example, in Figure 6b the estimate for clinical LDLc on Major adverse cardiovascular events appears to include the null after excluding individuals on statins, which I believe will surprise readers and could be attributed to this source of bias.

I believe this is a limitation for the resource developed by the authors as for example some readers may not realise the impact of medications on findings. For example if users attempt to query results from the web application for ApoB and ischemic heart disease (I25) (as I have just attempted - HR=0.92 :/) then it may lead them to cast doubt on what I believe is a valuable resource the community as long as this caveat is taken into consideration.

To address my only major concern, having given mitigating this issue some consideration in the past myself, would be provided estimates stratified by age group, for example as conducted by Bell et al (this papers ref) who stratified the UKB sample into tertiles. Given that age cannot be a collider between metabolites and disease outcomes, these age-stratified estimates can be used by readers to easily identify results in the full sample which may be prone to the influence of colliders (such as medication usage). Furthermore, estimates in the youngest tertile will be least prone to this source of bias given that a relatively small proportion of these participants will be taking statins.

I appreciate that this is a fairly large amount of work even if the authors have automated their analysis pipeline (which I expect they have given the huge amount of work already conducted in this study), however I do believe it will particularly important as there are readers who will no doubt be interested in using this resource to evaluate the relationship between lipoprotein lipid metabolites (which make up a lot of the 249 NMR traits) and cardiometabolic disease endpoints. It would be really valuable to the community to provide these age-stratified results via the web application, as well as emphasising these findings in the paper itself. This would be aided by an example such as how the age-stratified results vary between groups for the same lipoprotein metabolite-CVD disease relationship. With this addition I look really forward to using this resource in my future research (particularly as the web application is excellent) and thank the authors for putting it

together.

Reviewer #2 (Remarks to the Author):

This is a very nice and well written paper about the Nightingale Health dataset in UK Biobank. The authors provide a relatively brief overview of the dataset, summarize associations with disease endpoints, biomarker signatures, replication in a Finnish cohort and a public browser. Overall, I see no reason to suggest any major changes, and just have a 2 minor comments.

[1] Biomarker associations with disease could be a consequence of disease rather than a cause. I appreciate that "we focus on analyses of future onset of diseases from ICD-10 chapters" but this still may not completely imply causation. The people could still have the disease (or early stages) when blood was taken. Have you stratified by time between date of diagnosis and date of blood draw?

[2] Figure 2 : The $-\log_{10}(p)$ plot doesn't add much as points are not labelled. I would remove it and add p-values to the side of the hazard ratios.

Reviewer #3 (Remarks to the Author):

This paper presents results from a metabolomic analysis of a large sub-cohort of the UK Biobank. The Nightingale NMR spectroscopy platform is used to assay around 250 small molecule and lipoprotein measures from blood samples. This data is a welcome addition to the impressive arsenal built up by the biobank, representing around 20% of the whole cohort and the remaining 80% are to be completed in the coming years. I believe this paper and the accompanying 'atlas' resource is a very valuable addition to the literature. I have a few minor concerns detailed below.

- 1. The atlas allows public access to the biomarker-disease association results only. Please clarify for readers how they can access the individual level data.**
- 2. Please clarify why only 37 biomarkers are reported here and why only with diseases from ICD chapters A-N?**
- 3. What variables were adjusted for in the models? Nothing is stated in caption for the models of figure 2, but figure 3 indicates age, sex and assessment centre were used as covariates. Was this the case for all models?**
- 4. Fig 2d and line 159-160. According to the figure, alanine is significantly associated with many diseases, not only diabetes as stated in the text. For example gout, kidney disease, anemia etc.**
- 5. The fact that most biomarkers show a consistent direction of association with many diseases is interesting. Do the authors have any explanation for this?**
- 6. Discussion: the metabolite data show a surprisingly wide set of associations with a broad range of diseases. How does this compare to other omics data in the UK Biobank? Do other molecules show a similarly large range of associations across diseases? Also what % variance in the phenotype is accounted for. Metabolic biomarkers are often much more predictive than e.g. SNPs.**
- 7. Due to the wide range of associations, the biomarkers may not be very specific. Can the authors comment on the lack of specificity and implications for likely practical (e.g. diagnostic) use?**
- 8. L455 'log1p' Should this be log10?**

9. I am a little confused about which statistical models were used for which analysis. The methods say the authors used "Cox proportional hazard modelling to estimate associations between biomarkers and incident disease" and then states "For biomarker association testing with prevalent diseases, we used logistic regression". The authors should clarify which model was used for each of the results presented in the figures/tables. For example do the results in Fig 2 derive from Cox or logistic models?

10. Appendix 1: Outlier plates: where are these identified? This would be important for researchers reusing the data.

11. Appendix: Please report the correlation coefficients and regression parameters for all comparisons with routine clinical chemistry for the UK Biobank analysis. (Only 9 comparisons shown in the figure AF6).

REVIEWER COMMENTS

Reviewer #1 (Remarks to the Author):

This paper by Julkunen et al presents rich resource of associations between 249 circulating metabolic traits and over 700 ICD10 defined endpoints using data on 118,461 participants from the UK Biobank (UKB) study. As the authors highlight, this sample size provides a 10-fold increase in sample over previous profiling studies to date and further enhances the already amazing phenotypic breadth of the UKB.

This resource paper is an excellent contribution to the field with appropriate statistical analyses, clear presentation and summary of key findings/figures and an accompanying web resource to query/download results.

We thank the reviewer for their positive comments on the paper.

My only major concern is the inherent challenge of conducting NMR profiling in a biobank-scale dataset such as UKB and the most appropriate manner to account for medication usage (in particular statins). The authors touch on this on line 275:

'This observation, which is surprising compared to existing literature on LDL as a risk factor, is seen in both the NMR and clinical chemistry measurements, indicating that it stems from characteristics of the UK Biobank study rather than any property of the NMR measurements.'

The authors conduct further analyses excluding reported statin users from their sample in an attempt to correct for this source of bias, although I'm not sure this is appropriate given that such an approach may induce collider bias into their findings. For example, in Figure 6b the estimate for clinical LDLc on Major adverse cardiovascular events appears to include the null after excluding individuals on statins, which I believe will surprise readers and could be attributed to this source of bias.

We thank the reviewer for insightful comments regarding statin usage. We agree with the inherent challenge of appropriately accounting for medication usage and appreciate that excluding statin users might in some instances introduce collider bias into the results. To address this concern, we have now added the following analyses in the manuscript:

1. As suggested below, we have repeated analyses of all biomarkers vs all disease endpoints by age tertiles, with the youngest tertile having lower prevalence of statin use than the older age groups. We have included all these results in the atlas webtool as a resource for the research community. We have also included a new figure of the age stratified association results (Figure 7) and discuss the findings on p. 13.
2. To further examine whether statin use is the main cause of the inverse observations on LDL-related biomarkers in UK Biobank, we have also replicated the analyses in the FINRISK 1997

cohort, which collected blood samples in 1997, when statin use was not yet widespread (Supplementary Figure 9). The prevalence of statin use in this cohort is only 3.5% (4.5% after age matching to UK Biobank), which is even lower than the statin prevalence in the youngest tertile in UK biobank (6%).

The FINRISK 1997 replication results provide a helpful perspective for evaluating the results obtained in the statin-free population in UK Biobank, as these results are not subject to collider bias from the statin stratification. Many of the observed associations based on the statin-free subset in UK Biobank, including the null association of LDL-cholesterol with MACE, replicated in the FINRISK 1997 cohort, especially when matching to the age range in the UK Biobank. This suggests that the observations of the LDL biomarkers relate to the characteristics of the higher aged participants in UK Biobank rather than being introduced by collider bias from the statin user exclusion. We have added a paragraph to cover these findings on p. 13.

I believe this is a limitation for the resource developed by the authors as for example some readers may not realise the impact of medications on findings. For example if users attempt to query results from the web application for ApoB and ischemic heart disease (I25) (as I have just attempted - HR=0.92 :/) then it may lead them to cast doubt on what I believe is a valuable resource the community as long as this caveat is taken into consideration.

To address my only major concern, having given mitigating this issue some consideration in the past myself, would be provided estimates stratified by age group, for example as conducted by Bell et al (this papers ref) who stratified the UKB sample into tertiles. Given that age cannot be a collider between metabolites and disease outcomes, these age-stratified estimates can be used by readers to easily identify results in the full sample which may be prone to the influence of colliders (such as medication usage). Furthermore, estimates in the youngest tertile will be least prone to this source of bias given that a relatively small proportion of these participants will be taking statins.

I appreciate that this is a fairly large amount of work even if the authors have automated their analysis pipeline (which I expect they have given the huge amount of work already conducted in this study), however I do believe it will particularly important as there are readers who will no doubt be interested in using this resource to evaluate the relationship between lipoprotein lipid metabolites (which make up a lot of the 249 NMR traits) and cardiometabolic disease endpoints. It would be really valuable to the community to provide these age-stratified results via the web application, as well as emphasising these findings in the paper itself. This would be aided by an example such as how the age-stratified results vary between groups for the same lipoprotein metabolite-CVD disease relationship. With this addition I look really forward to using this resource in my future research (particularly as the web application is excellent) and thank the authors for putting it together.

We highly appreciate the suggestion and have now performed extensive analyses to generate the age stratified association estimates. We have added a new figure of the age stratified results (Figure 7) and cover these findings on p. 13. We also note that some biomarkers which are not influenced by statin use are stronger in the youngest age tertile, so it can be difficult to disentangle the impact of

potential collider bias from statins compared to other age-related effects. Nonetheless, we agree that these age stratified results can provide valuable information to the community and have now made them also available via the atlas web application. We expect that these results may facilitate future studies focusing on ageing and statin effects in relation to specific endpoints.

Reviewer #2 (Remarks to the Author):

This is a very nice and well written paper about the Nightingale Health dataset in UK Biobank. The authors provide a relatively brief overview of the dataset, summarize associations with disease endpoints, biomarker signatures, replication in a Finnish cohort and a public browser. Overall, I see no reason to suggest any major changes, and just have a 2 minor comments.

We thank the reviewer for their positive comments on the paper.

[1] Biomarker associations with disease could be a consequence of disease rather than a cause. I appreciate that "we focus on analyses of future onset of diseases from ICD-10 chapters" but this still may not completely imply causation. The people could still have the disease (or early stages) when blood was taken. Have you stratified by time between date of diagnosis and date of blood draw?

We certainly agree that some of the reported associations are likely to be a consequence of early disease process, and do not mean to imply that they are causal. To address this concern, we have now carried out sensitivity analyses by excluding the first two years of follow-up to ensure that the individuals were free of a clinical presentation of the disease at baseline (Supplementary Figure 2). The associations remained essentially unaltered because of this, noted now on p. 7.

[2] Figure 2 : The $-\log_{10}(p)$ plot doesn't add much as points are not labelled. I would remove it and add p-values to the side of the hazard ratios.

Thank you for the suggestion. We have now removed the Manhattan ($-\log_{10}(p)$) plots and added p-values along with the hazard ratios in Figure 2.

Reviewer 3

This paper presents results from a metabolomic analysis of a large sub-cohort of the UK Biobank. The Nightingale NMR spectroscopy platform is used to assay around 250 small molecule and lipoprotein measures from blood samples. This data is a welcome addition to the impressive arsenal built up by the biobank, representing around 20% of the whole cohort and the remaining 80% are to be completed in the coming years. I believe this paper and the accompanying 'atlas' resource is a very valuable addition to the literature. I have a few minor concerns detailed below.

We thank the reviewer for the positive evaluation of our work.

1. *The atlas allows public access to the biomarker-disease association results only. Please clarify for readers how they can access the individual level data.*

Instructions for accessing the individual level data are included in the Data Availability statement at the end of the Methods section (p. 22).

2. *Please clarify why only 37 biomarkers are reported here and why only with diseases from ICD chapters A-N?*

The 37 biomarkers have been clinically certified for diagnostics use as a part of the Nightingale Health NMR platform. Hence, in order to facilitate potential translational applications and visualization of the results for the examples highlighted in the paper, we focused on this set of 37 clinically validated biomarkers. These biomarkers also span most of the different metabolic pathways measured by the NMR platform. The majority of the remaining measures are very detailed lipoprotein measures. Complete results for all 249 biomarkers measured are available in the related atlas webtool. We have now clarified this in the Methods section (p. 18).

The disease selection of ICD10 chapters A-N was chosen to place the focus on common diseases. The selection of chapters A-N excludes pregnancy related outcomes, conditions originating in the perinatal period and congenital malformations, deformations and chromosomal abnormalities (chapters O-Q) as there were not enough incident events passing the criteria of over 50 events to be included in the analyses. Chapters R-U (including symptoms, signs and laboratory findings not elsewhere classified, injuries, accidents and factors influencing health status and contact with health care services and codes for special purposes) were excluded as we wanted to focus on common diseases in the present analyses. We have now clarified this in the Methods section (p. 19).

3. *What variables were adjusted for in the models? Nothing is stated in caption for the models of figure 2, but figure 3 indicates age, sex and assessment centre were used as covariates. Was this the case for all models?*

Correct, all models have been adjusted for age, sex and assessment centre. We have now clarified this in all figure captions.

4. *Fig 2d and line 159-160. According to the figure, alanine is significantly associated with many diseases, not only diabetes as stated in the text. For example gout, kidney disease, anemia etc.*

We do not mean to imply that alanine would only associate with diabetes. With the sentence being referred to: *“For instance, the amino acid alanine was almost exclusively associated with the risk of diabetes and related complications (Figure 2d).”*, we also intend to cover diabetes related complications. Most of the diseases in Figure 2d are either directly related to diabetes or are among common complications of diabetes (including for instance, kidney disease, heart disease, retinopathies and neuropathies). We have now rephrased the wording to clarify this point (p. 5).

5. *The fact that most biomarkers show a consistent direction of association with many diseases is interesting. Do the authors have any explanation for this?*

We agree that the observed similarity of biomarker association profiles across various types of diseases is interesting, and we touch upon this topic in Discussion (p. 15). This observation is also in line with the earlier studies that have linked plasma metabolites to multimorbidities (Pietzner et al. 2021; Kettunen et al. 2018) and all-cause mortality (Deelen et al. 2019), supporting the role of these biomarkers as shared risk markers across various types of health outcomes. As noted in the discussion, one potential explanation for the shared biomarker risk profiles could be that the biomarkers provide a reflection of general frailty, low-grade systemic inflammation and impaired immune response to various types of diseases. This would explain why many of the biomarkers have also been associated with susceptibility to severe infectious diseases (Julkunen et al. 2021) and all-cause mortality (Deelen et al. 2019). However, further research is required to elucidate the disease mechanisms and we hope that this resource can provide a valuable starting point for such detailed studies.

6. *Discussion: the metabolite data show a surprisingly wide set of associations with a broad range of diseases. How does this compare to other omics data in the UK Biobank? Do other molecules show a similarly large range of associations across diseases? Also what % variance in the phenotype is accounted for. Metabolic biomarkers are often much more predictive than e.g. SNPs.*

The broad range of associations across various diseases is certainly interesting, as also discussed in point 5 above. As noted on p. 10, the associations are of a similar magnitude to clinical biochemistry (Figure 5), which indeed is much higher than what is reported e.g. with individual SNPs. Some protein markers available in UK Biobank (e.g. C-reactive protein and cystatin C) display a similar wide range of disease associations as noted in the Discussion to be known for inflammatory biomarkers. In contrast, other proteins (e.g. rheumatoid factor) have specific associations with only certain diseases. The wide span of associations of the metabolic biomarkers contrasts the lessons from polygenic risk scores for which the associations generally become stronger the more the specific the disease outcome is. Computing the percentage of variance explained is not straightforward with this type of censored survival data, where the binary disease outcome definition is dependent on the length and availability of the follow-up time. We report all association magnitudes in SD units to facilitate comparison between cohort studies and to other types of data available in the UK Biobank, but leave such comparisons outside the scope of the present study.

7. *Due to the wide range of associations, the biomarkers may not be very specific. Can the authors comment on the lack of specificity and implications for likely practical (e.g. diagnostic) use?*

We certainly agree that the similarity of the biomarker risk profiles can pose challenges for certain clinical applications requiring high disease specificity. On the other hand, the lack of specificity can be advantageous when aiming to use the biomarker panel to assess the risk of

multiple diseases and overall health status simultaneously based on a single measurement. For instance, this could potentially be used for individualized health checkups at scale to prioritize high-risk individuals for further examinations and guide preventative actions. In fact, a study recently published in Nature Medicine (Bruegel et al. 2022) demonstrated the potential of the NMR biomarker profiles to predict multidisease outcomes in UK Biobank, showing predictive improvements over comprehensive clinical risk factors which were shown to largely translate into clinical utility. As such, this could have many applications in clinical settings and provide an attractive tool for multi-disease risk screening, despite the individual disease specificity being limited in some instances. We have now added a paragraph in the discussion (pp. 15-16) to discuss these implications for clinical use.

8. *L455 'log1p' Should this be log10?*

We do intend to say "log1p", which is the term used for logarithm of 1 + x. We have now clarified this term in the Methods section (p. 18).

9. *I am a little confused about which statistical models were used for which analysis. The methods say the authors used "Cox proportional hazard modelling to estimate associations between biomarkers and incident disease" and then states "For biomarker association testing with prevalent diseases, we used logistic regression". The authors should clarify which model was used for each of the results presented in the figures/tables. For example do the results in Fig 2 derive from Cox or logistic models?*

Thank you for pointing this out. As explained in the main text (p. 5) and in the Methods section, the examples highlighted in the paper focus on the analysis of future incidence of diseases, in which case the Cox model was always used. However, as the biomarker atlas webtool also includes associations with prevalent disease, we have included a description of the statistical methods used for the analyses of prevalent diseases in the Methods section. We have now clarified this in all figure and table captions, along with the information about the covariates used for adjusting the models as pointed out in comment 3).

10. *Appendix 1: Outlier plates: where are these identified? This would be important for researchers reusing the data.*

In the Appendix, we point out that an independent study from Cambridge university (Ritchie et al. 2021) has discovered a small number of these outlier plates (affecting on average less than 1% of the samples), with deviations seemingly arising from UK Biobank's sample plating process. We have now clarified the reference to this study and the R package they have developed for removing the technical variation, including the outlier plates (p. 6 in the Appendix). Since the outlier plates constitutes such a small fraction of the overall samples, the results of disease-association testing are virtually unaltered by removal of these few outlier plates. Therefore, we did not do this exclusion for the analyses presented in the present manuscript.

11. Appendix: Please report the correlation coefficients and regression parameters for all comparisons with routine clinical chemistry for the UK Biobank analysis. (Only 9 comparisons shown in the figure AF6).

The nine comparisons shown in Figure AF6 already include all biomarkers that overlap between the NMR platform and clinical chemistry. Naturally, we can only show these comparisons for biomarkers that are measured by both platforms. We have now clarified in the Figure AF6 caption that these nine biomarkers cover all the overlapping biomarkers between the two platforms.

REVIEWERS' COMMENTS

Reviewer #1 (Remarks to the Author):

I thank the authors for addressing my comments. I believe this resource will be extremely helpful to the field and I look forward to citing it in my future research.

Reviewer #2 (Remarks to the Author):

The authors have addressed my comments

Reviewer #3 (Remarks to the Author):

The authors have addressed all my concerns. Thank you.

REVIEWERS' COMMENTS

Reviewer #1 (Remarks to the Author):

I thank the authors for addressing my comments. I believe this resource will be extremely helpful to the field and I look forward to citing it in my future research.

We thank the reviewer for the positive feedback.

Reviewer #2 (Remarks to the Author):

The authors have addressed my comments

We thank the reviewer for the comments.

Reviewer #3 (Remarks to the Author):

The authors have addressed all my concerns. Thank you.

We thank the reviewer for the positive evaluation of our work.